# Transcriptional Basis of Psoriasis from Large Scale Gene Expression Studies: The Importance of Moving towards a Precision Medicine Approach

**DOI:** 10.3390/ijms23116130

**Published:** 2022-05-30

**Authors:** Vidya S. Krishnan, Sulev Kõks

**Affiliations:** 1Centre for Molecular Medicine and Innovative Therapeutics, Murdoch University, Discovery Way, Murdoch, WA 6150, Australia; vidya.saraswathykrishnan@murdoch.edu.au; 2Perron Institute for Neurological and Translational Science, 8 Verdun Street, Nedlands, WA 6009, Australia

**Keywords:** alternative transcripts, psoriasis, atopic dermatitis, transcriptomics, S100 proteins, keratinocytes

## Abstract

Transcriptome profiling techniques, such as microarrays and RNA sequencing (RNA-seq), are valuable tools for deciphering the regulatory network underlying psoriasis and have revealed large number of differentially expressed genes in lesional and non-lesional skin. Such approaches provide a more precise measurement of transcript levels and their isoforms than any other methods. Large cohort transcriptomic analyses have greatly improved our understanding of the physiological and molecular mechanisms underlying disease pathogenesis and progression. Here, we mostly review the findings of some important large scale psoriatic transcriptomic studies, and the benefits of such studies in elucidating potential therapeutic targets and biomarkers for psoriasis treatment. We also emphasised the importance of looking into the alternatively spliced RNA isoforms/transcripts in psoriasis, rather than focussing only on the gene-level annotation. The neutrophil and blood transcriptome signature in psoriasis is also briefly reviewed, as it provides the immune status information of patients and is a less invasive platform. The application of precision medicine in current management of psoriasis, by combining transcriptomic data, improves the clinical response outcome in individual patients. Drugs tailored to individual patient’s genetic profile will greatly improve patient outcome and cost savings for the healthcare system.

## 1. Introduction

Psoriasis is a chronic autoimmune skin disease with a complex etiology involving various genetic and environmental factors. It is characterized by inflammation and altered epidermal differentiation, inflammatory cell infiltrate in the epidermis and dermis, and dilation of superficial dermal blood vessels leading to red lesional plaques and scaling. The prevalence of psoriasis on a global scale across countries from published studies varies between 0.09% and 11.4%, but in most developed countries, the prevalence ranges from 1.5% to 5% [1,2]. Almost 90% of individuals with psoriasis have the most common form of the disease, known as psoriasis vulgaris or plaque psoriasis, characterized by well-demarcated, erythematous plaques with overlying, coarse scale. Other major subtypes of psoriasis include guttate psoriasis, which typically presents as the acute onset of numerous small, inflammatory plaques; pustular psoriasis, which may present as an acute, subacute, or chronic pustular eruption; and erythrodermic psoriasis, which exhibits cutaneous erythema and scale involving most or all the body surface area. There is growing and emerging evidence that psoriasis patients have a higher prevalence of associated comorbid disease multiple comorbidities, including psoriatic arthritis, cardiovascular disease (CVD), metabolic syndrome, mental health conditions such as depression and anxiety, inflammatory bowel disease (IBD), and malignancies such as lymphoma [1,3]. 

Although psoriasis can affect all ages, early onset for females and males occur around the age of 16 and 22, respectively, and late onset is mostly around 60 years for females, compared to 57 years for males [4]. Family-based and population-based epidemiological studies support a clear genetic contribution to the development of psoriasis. Having a family history increases the risk of psoriasis. Studies on monozygotic twins have reported a 70% chance of one twin developing psoriasis compared to 20% for paternal twins [5,6]. PSOR1 is the major susceptibility locus for psoriasis, encoding the gene variant HLA-Cw6. A majority of early onset patients (around 85%) carries this locus, compared to late onset patients (around 15%) [4,7,8]. HLA-Cw6 is associated with various aspects of psoriasis including genetic susceptibility, clinical manifestation, comorbidity, and treatment efficacy [8]. Around nine psoriatic susceptibility loci have been identified by genome-wide association studies (GWAS), revealing several genomic markers [9,10,11,12]. From a large meta-analysis of GWAS, with a sample data of >39,000 individuals (including data from eight Caucasian cohorts), Tsoi et al. identified 16 additional psoriasis susceptibility loci for psoriasis. This study increased the number of identified loci to 63 for European-origin individuals and also highlighted the role of NFκB and interferon pathways, thereby providing insights to the pathogenesis of psoriasis [13,14]. Other studies have also mapped psoriasis susceptibility loci, including PSORS1 on 6p21.3, PSORS2 on 17q, PSORS3 on 4q, PSORS4 on 1cen-q21, PSORS5 on 3q21, and PSORS6 on 19p13 [15].

Transcriptome profiling and gene expression studies in psoriasis can provides valuable insights to the molecular basis of the disease, detecting new biomarkers and therapeutic targets. Microarray and RNA sequencing are relevant tools to detect new mRNA transcripts and alternatively spliced forms [7,16]. Most of the current disease management approaches for psoriasis and psoriatic arthritis (PsA) depends on clinical assessments, and are often not precise. Hence, a more precise approach, which takes into account an individual’s variations in genes, proteins, environment, and lifestyle, will provide better insight to help design precise therapies, leading to better treatment outcome. This comprehensive review focused on the psoriasis transcriptome, with particular emphasis on the differentially regulated transcripts and the expression profile of alternatively spliced transcripts and its importance as potential therapeutic targets in the disease treatment. 

## 2. Large Cohort Transcriptome Analysis of Psoriasis 

The skin epidermal cells/keratinocytes are the main players contributing to the characteristic epidermal hyperplasia, immune response activation and recruitment of inflammatory cells, which are characteristic of the early onset of psoriasis pathogenesis [17]. A well-regulated balance of cell renewal and cell differentiation processes is essential for epidermal homeostasis [18]. This balance is impaired in chronic inflammatory diseases, such as psoriasis, leading to abnormal differentiation of keratinocytes [19]. Significant proliferation (around 80%) of keratinocytes, along with immune cell infiltration (including T-cells and macrophages) is a characteristic feature of psoriatic lesions. Induction of innate immunity pathways is one of the early onset features in psoriasis [16,20]. All this emphasises the strong immunological component associated with its disease onset. This chronic immunological dysfunction in psoriasis is driven by immune cells, including Th1 and Th17 cells, as well as pro-inflammatory cytokines, primarily interleukin-17 (IL-17), interleukin-23 (IL-23), interleukin 22 (IL-22), tumour necrosis factor (TNF), and interferon-γ [21,22]. IL-17/IL-22-producing T cells are mainly associated with the initial phases of psoriasis, while IFN-producing Th1 and Tc1 cells mostly regulate the chronic phase of the disease. In addition to the IL17/IL22/TNF-α/IL-23 cytokine pathway, focus has recently shifted to the interleukin-1 (IL-1) family of cytokines, which includes interleukin-36 (IL-36) cytokine and IL-36 receptor antagonist (IL-36Ra/IL36RN) [23,24]. IL-36 cytokines are mainly expressed in epithelial and immune cells, and include three agonistic pro-inflammatory cytokines (IL-36α, IL-36β and IL-36γ) and two anti-inflammatory cytokines that function as antagonists (IL-36RN or IL-36Ra and interleukinL-38). All these cytokines share a common a receptor, complex IL-36R [25,26]. Skin and serum samples of psoriatic patients reveal overexpression of IL-36α, IL-36β, and IL-36γ, showing a strong correlation between disease severity and cytokine levels [27]. Whole transcriptome analysis of lesional (LP) and non-lesional psoriasis (NLP) skin samples, in comparison with healthy control samples, found that, along with an up-regulated IL17/IL22 cytokine network, IL36G and IL36RN were also identified as highly expressed in psoriatic lesions, with LP samples showing much stronger expression compared to NLP samples [16].

The epidermal differentiation complex (EDC) in in the chromosome 1q21 region encodes the major genes involved in epidermal differentiation [28]. These genes constitute three main families (a) the cornified envelope precursor proteins (loricrin, involucrin, and the family of late cornified envelope proteins), (b) the keratin filament-associated proteins (filaggrin, trichohyalin, repetin, hornerin, and cornulin), and (c) the S100 calcium-binding proteins; importantly, S100A7, S100A7A, S100A8, S100A9, and S100A12 [8,29,30]. Since altered epidermal differentiation is an important feature of psoriatic skin pathogenesis, these EDC genes are one of the most important psoriatic markers/gene candidates. 

Large scale gene expression studies help us to gain insight to the molecular basis underlying the disease pathogenesis and progression of psoriasis. Microarray-based studies of the psoriatic transcriptome have revealed a large number of differentially expressed genes (DEGs) in lesional and non-lesional skin [31]. The first longitudinal study was undertaken by Oestreicher et al. to understand how expression levels of a set of 159 genes and their transcripts were differentially expressed in lesional and psoriatic skin. It was found that most of these genes were involved in different aspects of gene regulation including intracellular signalling, cell cycle, transcriptional, and translational regulation [15]. A similar array-based gene expression studies, comparing 15 psoriasis patients and healthy controls, by Bowcock et al. identified a total of 177 genes differentially expressed in lesional skin versus normal skin [32]. One of the first gene expression studies that compared skin lesions of atopic dermatitis (AD) and psoriasis reported a significant upregulation of 62 genes, including CCL4, CCL20, CXCL2, CXCL8, and CXCR2, in the psoriatic skin, while around 18 genes were upregulated in AD skin [33]. Overlapping of around 2000 genes (including CCL-2/3/17/18, IL-6/8/17A/22/23A, S100A9/A15, TRPV1, and PLA2) unique to psoriatic, lesional atopic skin, compared with healthy skin, were identified from a recent RNA-seq analysis from 25 psoriatic and AD skin patients [34]. Such studies provide more insights to the characteristic disease signature of both these skin diseases [34]. 

Several lines of evidence suggests that keratinocytes (KC) are the main drivers in the development of psoriasis in response to activated immune cascade [7,8]. Studies by Zhou et al. and Kulski et al. have identified that many of the genes upregulated in psoriatic skin were involved on epidermal organization and differentiation, highlighting the role of keratinocytes in the pathogenesis of psoriasis [35,36]. Several in vitro studies based on cytokine stimulation of keratinocytes have also provided insights to the cellular sources of the gene expression in psoriasis [20,37,38,39,40]. Establishing an in vitro psoriasis KC model by stimulating Hacat keratinocytes with an M5 cytokine cocktail (IL17A, TNFα, Oncostatin, IL22, and IL1α) resulted in strong transcriptional regulation of chemokines, including CXCL1, CXCL2, CXCL8, CCL20, and CCL27, as well as antimicrobial peptides S100A7, S100A8, S100A9, S100A12, LL-37, and Beta Defensin-2 [37]. However, one of the main limitations of such in vitro models, involving monolayer keratinocytes, is that they may not fully recapitulate psoriatic features present in psoriatic lesions and uninvolved psoriatic skin. One early study comprising a large cohort of lesional vs. non-lesional transcriptomic profiling was performed by Gudjonsson et al., in which they analysed 58 paired samples of lesional and non-lesional skin in comparison with 64 control biopsies [41]. Some of the strongly upregulated genes included SERPINB4, PI3, DEFB4, and several S100 family members. This study also reported significant dysregulation of genes involved in lipid and fatty acid metabolism, including transcription factors PPARA, ESR2, and SEBF1, linked to impaired lipid metabolism. An interesting observation noted in this study was that uninvolved psoriatic skin often exists in a “pre-psoriatic” state. When compared to already reported/published transcriptomes of cytokine stimulated cultured keratinocytes, there was little overlap in the gene expression data with the lesional psoriatic dysregulated transcriptome. [20,37,38,39]. A very recent study by Pasquali et al. studied the keratinocyte-specific gene expression changes in lesional and non-lesional psoriasis skin, and the results presented in this study show evidence for the dominance of an IL-22/IL-17A signature in psoriatic keratinocytes with the contribution of the IL-1/IL-36 and IL-20 families [42]. Molecular heterogeneity between various subtypes of plaque psoriasis was studied by Ahn et al. on skin samples from individuals with scalp, palmoplantar, and conventional plaque psoriasis compared with healthy controls. Around 763 differentially expressed genes, including S100A7A, S100A9, SERPINB4, KRT6, SPRR2A/B, C10orf99, Il36γ, and late cornified envelope proteins associated with keratinocyte proliferation and activation, were reported. [39]. Studies have investigated RNA expression patterns across lesional and non-lesional tissue samples vs. healthy controls. Decision tree predictors to differentiate psoriatic samples based on gene expression patterns were studied by Ainali et al., which revealed distinct molecular sub-groups of plaque psoriasis. Enrichment analysis revealed two networks PP01 and PP02 linked to different biochemical pathways for the two lesional psoriatic subgroups. PP01 network cluster was associated with Notch, Wnt, TGFβ, and ERbB signalling pathways, while PP02 was enriched with metabolic pathways. This suggested that therapies targeting these pathways could be employed for the respective subgroups [43]. One large scale study, involving 85 matched pairs of lesional and non-lesional biopsies from patients, was performed by Suarez Farinas et al. and identified 2725 differentially expressed genes in psoriatic plaques. In agreement with many previous studies, S100 family of proteins and SERPINs were the strongly upregulated genes, along with many peptidases, including Kallikrein-related peptidase-6 (KLK6) and kallikrein-related peptidase-13 (KLK13) [44,45,46]. Many genes associated with lipid and fatty acid metabolism, as well as chemokines CCL27, were down regulated in these plaques [44]. A potential linkage between psoriatic skin gene expression and co-morbidities was reported in patients with moderate-to-severe psoriatic compared with healthy controls. Serum profiles identified strong association between functional pathways in lesional skin with diabetes and cardiovascular risk pathways [46]. Several new DEGs, not reported in previous microarray studies, were found from another large scale RNA-seq analysis from lesional skin biopsies from 174 individuals [31]. This comprehensive study by Li et al. made an interesting observation, that, although RNA-seq and microarray measurements were consistent for intermediate and high abundance transcripts, several discrepancies were noticed for low abundance transcripts in both disease groups cases controls. The gene fold change (FC) also showed wider estimates for low abundance transcripts [31]. 

Interferons (IFNs) are one of the key players in psoriasis pathogenesis and have been found in psoriatic lesions, emphasising the importance of these cytokines in this disease [47,48]. Whole genome array analysis from paired lesional and non-lesional psoriatic skin have demonstrated that type I IFNs and type II IFN–inducible genes are strongly expressed in psoriatic skin. The mRNAs of type I IFN family members-IFNAR1, and IFNAR2 are upregulated in lesional skin, but not in non-lesional skin (except IFN-a5 and IFN-k). A significant expression of type 1 IFN-inducible genes, along with IFN-γ (which is a Type II IFN) and TNF-α in lesional skin (but not in non-lesional skin), was also observed in psoriatic patients examined in this study [47]. The upregulation of mRNAs of type I IFNs, IFN-γ, and TNF-α gene signatures in lesional skin suggests the presence of these cytokines and their active signalling in psoriasis, making it a potential therapeutic target for psoriasis treatment. In contrast, the elevated expression of mRNAs of IFN-γ or TNF-α did not correspond to upregulation of IFN-γ or TNF-α-inducible genes, suggesting their absence in non-lesional skin, or that other signalling molecules might have inhibitory effects on the IFN-γ and TNF-α pathways in non-lesional skin of psoriatic patients [47]. A separate study by one of the above mentioned authors found that a single injection of IFN-γ to the dermis of non-lesional sites of psoriasis can recapitulate psoriasiform immune response and the transcriptional profile changes in psoriatic plaques [45]. Similar influx was also observed in healthy, non-psoriatic individuals [49]. Great variability in treatment response exists between individual psoriatic patients, reflecting the heterogeneity of various inflammatory networks driving the disease. Psoriasis lesions looks clinically similar, with their characteristic pathophysical features, with standard immunohistochemistry unable to differentiate between lesional skin samples. It is now possible to identify such variability using whole genome transcriptomic profiling. One such study, involving 62 lesional skin samples from patients with stable chronic plaque psoriasis, identified and grouped lesions based on inflammatory gene expression signatures as strong (37%), moderate (39%), or weak inflammatory infiltrates (24%) [50]. Such studies play an important role in shaping treatment responses and catering personalized options for individuals [50]. In one large scale transcriptomic study by the same authors, involving 163 biopsies from psoriatic lesions, it was found that most of the DEGs were attributed to activated keratinocytes (56%), followed by T-cell infiltration (14%) and macrophages (11%). The decreased DEGs were mostly associated with adipose tissue and dermis [45,51]. 

As discussed above, many studies have reported consistent differences between lesional and non-lesional skin, emphasizing analyses of differentially expressed genes (DEGs) [44,52]. Availability of a larger cohort not only provides information underlying different disease mechanisms, but also allows us to detect gene variations and discern the molecular subtypes of the disease. This facilitates the development of new expression-based biomarkers and therapeutic targets for clinical applications [31,51]. Here, we have attempted to summarize the findings of several important large scale transcriptome studies and emphasise the importance of such studies to elucidate potential therapeutic targets for psoriasis treatment. A summary of findings from the selected large scale transcriptomic analyses have been summarised in Table 1. 

To address the variations in the expression of DEGs across different microarray experiments, it is important to have a statistically based meta-approach which can combine the results of various individual data. Such approaches provide a more accurate representation of a disease group than individual study data. A similar approach was used by Tian et al. who conducted a meta-analysis on five microarray data sets. This large scale study, including 193 LS and NL pairs, was termed the Meta-Analysis Derived (MAD) transcriptome [53]. The top genes identified in the MAD transcriptome were associated with atherosclerosis signalling and fatty acid metabolism, along with several ‘‘new’’ genes involved in cardiovascular development and lipid metabolism. These findings highlight the relationship between psoriasis and systemic co-morbidities, such as CVD and metabolic syndromes. Several psoriatic DEGs were reported in this transcriptome, which are potential targets for therapeutic treatments and further research [53]. A MAD transcriptome was also established for AD by the same team by combining four published AD datasets to define robust disease profile for AD and was termed the meta-analysis derived AD (MADAD) transcriptome [54]. Genes involved in lipid metabolism were identified, which included the fatty acid 2 hydroxylase (FA2H) protein, essential to the de novo synthesis of specific ceramides that are critical in maintaining the permeability barrier of the epidermis and ELOVL fatty acid elongase 3 (ELOVL3), which is essential for the prevention of trans-epidermal water loss [54]. A graphic model showing molecular changes in psoriasis is shown in Figure 1. 

A transcriptome-wide association study (TWAS) based on results from GWAS of psoriasis (5175 cases and 447,089 controls) and gene expression levels from six tissues datasets (blood and skin) revealed novel candidate susceptibility genes of psoriasis [55]. One of the important genes identified was the EFEMP2 gene, associated with cardiovascular disease, which is a frequent co-morbidity of psoriasis. Mice lacking the EFEMP2 gene, display increased aorta wall thickness, abnormal smooth muscle morphology, and aorta aneurysm. 

The histopathological hallmarks of psoriasis typically include hyperkeratosis with parakeratosis, immune cell infiltration, acanthosis thickening, abscess formation, and vascular dilatation congestion [56]. However, the genetic factors causing these features are largely overlooked. To identify genes that contribute to or regulate psoriasis-specific histopathological features, a binding and expression target analysis (BETA) was performed on a cohort of 60 skin biopsies (including 20 lesional, 20 non-lesional, and 20 controls) [57]. Several upregulated genes were linked with chromatin accessibility, and it was found that transcription factor AP-1 regulated increased expression of some of these genes (SQLE, STRN, E1F4, and MYO1B). Increased chromatin accessibility facilitated the binding of AP-1 to these target genes and induced their expression. This increased expression of these genes correlated with hyperkeratosis, parakeratosis, and acanthosis thickening in patients [57].

## 3. Alternative Splicing Variants: Psoriasis Isoforms

Alternative splicing of precursor mRNA is a fundamental regulatory mechanism of gene expression [58]. Pre-mRNA, which has introns removed and exons joined at different combinations, generates new alternative transcript isoforms. Thus, different transcript isoforms produced from a single gene lead to the production of several protein variants, which impacts gene function through various mechanisms [59]. Alternative splicing not only provides insights into fundamental gene expression, but can also be harnessed as a strategy for therapeutic interventions [58]. It is estimated that more than 95% of genes may undergo alternative splicing [60,61,62].

The mechanism of alternative splicing in psoriasis has been largely overlooked. A recent study by Li et al. looked into the potential role of splicing in psoriasis using mouse and human data sets [63]. This large-scale computational analysis looked into the RNA- seq data of altered splicing factors in human and mouse data sets, and found 18 conserved exon skipping (ES) splicing events in psoriasis, along with several candidate splicing factors that may regulate splicing in psoriasis. Many differential alternative splicing events were detected using psoriasis mouse and human datasets, and it was found that many exon-skipping events were conserved in humans and mice. Additionally, using splicing signature comparison analysis and their curated splicing factor perturbation RNA-Seq database, SFMetaDB, using the psoriasis datasets, nine candidate splicing factors that may be important in regulating splicing in the psoriasis mouse model dataset were identified, of which three were confirmed upon analysing the human data [63].

Psoriasis is characterized by significant differences in the expression of RNA alternative isoforms and gaining insight to these new isoforms is essential to design precise therapies for the treatment of psoriasis leading to a better outcome. Although several studies have been published that investigated potential links between transcriptome changes and psoriasis using RNA-seq and microarrays, few studies have analysed expression profile of alternatively spliced transcripts in psoriasis. One such study, by Koks et al. identified potential alternatively spliced RNA isoforms with disease specific expression profiles from lesional psoriatic (LP), non-lesional psoriatic (NLP), and normal skin using transcript-based annotation [64]. This enabled analyses of 173,446 transcripts, and around 9000 transcripts were identified as differentially expressed between study groups. RNA isoform-based analysis allows the detection of multiple transcripts for a gene, such as in the case of KLK10, two isoforms of which were significantly upregulated in the lesional skin [64]. Similar isoform complexity has also been reported for described for the S100 genes, a multigenic family of low molecular weight calcium binding proteins that play a significant role in mediating innate and acquired immune responses. These damage-associated molecular pattern (DAMP) molecules are often released under stress and inflammatory conditions, especially psoriasis [29]. Around 21 S100 proteins have been identified, out of which around 11 (S100A2, S100A3, S100A4, S100A6, S100A7, S100A8, S100A9, S100A10, S100A11, S100A12, S100A15, S100B, and S100P) are expressed in the normal and/or diseased epidermis [30]. The S100A7A/A15 (Koebnerisin) protein is overexpressed in psoriasis and is one of the most important markers in psoriasis [65,66].

The human S100A7A/A15 gene has two alternatively spliced mRNA isoforms, a short (0.5 kb), S100A15-S, and a long (4.4 kb), S100A15-L [66]. The long isoform is specific to lesional psoriatic skin, where it is highly upregulated compared to non-lesional skin [64]. S100A7A long form is not expressed in normal skin, while short is weakly expressed in normal epidermis [65,67]. The expression levels of these two isoforms indicates differential transcription stability for the hS100A15 mRNA isoforms in psoriasis and chronic atopic [65]. Thus, S100A7A is good example for the transcript-specific regulation of inflammation and disease progression [64]. Another example is S100A8, which is also a well-known psoriasis-inducing gene belonging to the same family of genes [68]. Two splicing variants of S100A8 (ENST00000368733 and ENST00000368732) were significantly upregulated in lesional skin [64]. Another splicing variant upregulated in lesional skin samples was the IL36RN gene, belonging to the IL-1 cytokine family.

Caspase recruitment domain family member 14 (CARD14) is an activator of NFκB within the epidermis. Alternative splicing of CARMA2/CARD14 transcripts generates protein variants with differential effect on NFκB activation [69]. Jordan et al. identified 15 missense variants in CARD14 in seven cohorts (including 6000 cases and 4000 controls). More variants were found in cases compared to controls, and these variants increased transcriptional activation of NFκB and enhanced expression of psoriasis specific transcripts [70,71].

TRAF3 interacting protein 3 (TRAF3IP2) mutations are often associated with psoriasis and PsA. The TRAF3IP2 gene encodes for proteins involved in I-17 signalling. It is reported that the exon-2-excluded isoform of TRAF3IP2, by alternative splicing, negatively regulates IL-17 signalling, resulting in overproduction of cytokines IL-17 and IL-22 leading to perturbed inflammatory cascade [11,63,72].

The expression of fibronectin (FN) isoforms, including extra domain A (EDA), has been reported to be overexpressed in psoriatic uninvolved epidermis sensitizes keratinocytes to mitogenic signals [73]. Hacats—immortalized keratinocyte cell lines—expressed a higher ratio of EDA^+^ mRNA and EDA^+^ isoform of fibronectin [74].

Hence, it is important to investigate splicing variants of a gene and its gene expression, as compared to single gene analysis, to identify potential new candidates for disease treatment of psoriasis. When combined with relevant tools, such as RNA-seq, this provides more information to design individualised treatment options for a better outcome of the disease.

## 4. Neutrophils and Blood Transcriptome Signature in Psoriasis

Lesional skin biopsies are usually the ideal sample source to investigate psoriasis pathogenesis. Although less relevant to psoriasis, sampling blood has the advantage, over others, of allowing repetitive sampling with minimal risk or discomfort. Blood samples also provide information regarding the immune status of the patients. This blood transcriptome profiling offers the advantage of providing all such relevant information for not only psoriasis, but also other inflammatory disorders [75]. Most existing transcriptomic studies on psoriasis pathogenesis have focused on comparing skin biopsies from psoriasis patients and normal subjects [13,31,42]. However, there are few studies that have examined transcriptome changes in the blood.

A significant increase of neutrophil numbers in psoriatic lesions is one of the histopathological hallmarks in psoriasis pathogenesis. The role played by neutrophils in psoriasis pathogenesis has received particular attention over recent years [75,76,77]. Evidence of neutrophil infiltration in lesional skin was earlier reported by Yao et al. [47]. In the early phase of plaque formation, neutrophils infiltrate the dermis of the skin, then migrate to the epidermis and stratum corneum, where they accumulate as pustules or microabscesses [78,79]. Blood transcriptome profiling from two public psoriasis data sets was studied by Rawat et al. [75]. This study found that there was a strong neutrophil-driven inflammation associated with blood transcripts of psoriasis patients. Such studies widen the possibility of assessing systemic inflammation in psoriasis. Moreover, blood transcripts are also relevant to assess markers of cardiovascular risk in psoriasis patients, since CVD is a common co-morbidity associated with psoriasis [75].

Peripheral blood of psoriatic patients exhibit two unique subsets of neutrophils (CD10_pos_ and CD10_neg_) at different maturation stages. The number of CD10_neg_ aged neutrophils was higher compared to number of CD_pos_ neutrophils in psoriatic skin, since healthy skin is reported to have a smaller number of aged neutrophils [78]. These neutrophils are known to induce IL-17 production in T-cells in vitro, and, upon antibody treatment, the number decreased significantly. This highlights the regulatory effect of neutrophils in psoriatic skin [78]. The role of peripheral blood mononuclear cells (PBMCs) in psoriatic inflammation was studied by Wang et al., who performed blood transcriptome analysis on samples from generalized pustular psoriasis (GPP) patients before and after acitretin treatment. It was found that low-density granulocytes accumulated in the PBMCs of psoriasis patients, and several genes involved in neutrophil recruitment and pattern recognition (FPR1, FPR2, etc.) were significantly down regulated [80]. Other downregulated transcripts included antimicrobial peptides (AMPs) S100A8, S100A9, and S100A12. All these results suggest the role of neutrophil activity in alleviation of GPP.

A positive correlation between inflammasome signalling and psoriasis disease severity was investigated by Garshik et al. using blood transcriptome profiling from a cohort of 20 patients and 10 age-matched healthy controls [81]. Upregulation of inflammatory genes, including IL-1β, suggested a linear relationship between differentially expressed inflammatory transcripts and disease severity. Endothelial transcriptome was assessed by collecting endothelial cells from psoriasis patients and healthy controls. This revealed significant upregulation of inflammatory transcripts, including IL-1β, CXCL1, CXCL10, VCAM-1, and CCL3. Taken together, this data indicates the link between psoriasis, inflammasome signalling, and impaired vascular health [81].

Recently, the same authors characterized the levels of PCSK9 and cardiovascular risk in the blood and skin of psoriatic patients [82]. The study, involving two separate human psoriasis cohorts, showed increased levels of PCSK9 in psoriatic patients compared to age-, sex-, and cholesterol-matched controls [82]. The PCSK9 levels correlated with impaired endothelial vascular health and coronary artery calcium score, suggesting an association between circulating PCSK9 and early, as well as advanced, stages of atherosclerosis in psoriasis. Krahel et al. demonstrated significantly elevated levels of serum concentrations of PCSK9 in patients with mild to moderate psoriasis, suggesting impaired lipid metabolism in psoriasis [83]. Serum PCSK9 levels had a strong correlation to BMI and triglyceride levels, indicating that PCSK9 could be a novel marker for psoriasis and associated cardiovascular risks in psoriatic patients. The authors also showed that three months of monotherapy with methotrexate greatly reduced the levels of PCSK9 in the blood, indicating that methotrexate could be a treatment choice in such patients.

Differentially expressed circular RNAs (circRNAs), long non-coding RNAs (lncRNAs), and mRNAs were reported in a whole blood analysis from patients with blood heat psoriasis [84]. Using a high-through microarray, it was revealed that a total of 205 circRNAs, 393 lncRNAs, and 157 mRNAs were differentially expressed, and these were associated with lipid metabolism, autoimmune pathways, and signal transductions. In an Affymetrix HG-U95A microarray study that compared nine sets of previously reported microarray data, 20 skin- and two blood-associated hot spots were identified, with 34.5% of genes overlapping from these multiple studies [85]. Most of the blood-related DEGs were associated with lipid and fatty acid metabolism, immune function, and proteolysis [85]. Very recently, the role of gut microbiome and host microbe associations in psoriasis was investigated by Chang et al. in a cohort of 26 psoriasis patients and 16 healthy controls. Diversity in gut microbiome was reported in psoriasis patients, and this gut flora was previously reported to be associated with several other autoimmune diseases [86]. Additionally, blood transcriptome analysis revealed higher numbers of activated CD4+ effector T cells and CD8+ T cells in psoriasis blood compared to healthy controls [86].

However, it should be noted that blood transcriptomic profiling has its inherent limitations. While systemic inflammation and interferon responses can be measured in whole blood, an immune response might not be possible, because such changes are mostly observed in affected skin.

## 5. Precision Medicine in Psoriasis Management

One of the main drawbacks of current management of psoriasis is the inability to predict treatment responses of psoriasis biologics in individual patients. An “assessment gap” is often observed between the moment the response to a treatment is assessed biologically and when it is determined clinically [87]. An “assessment gap” is an important part in the drug administration which predicts the time taken for the drug to reach its target and act through the signalling pathways eliciting its effect or failing to alter the pathogenic state. A personalized/precise approach, considering an individual’s genetic, cellular, and molecular variability, is essential for psoriasis disease management. The vast amount of transcriptomic information available for psoriasis makes it an excellent example of precision medicine, elucidating treatment outcome from molecular data. By combining the data and information generated by transcriptomic and other analyses, it is now possible to predict the effects and responses of drugs to provide treatments and therapies with high efficacy, thereby reducing the “assessment gap”. Such a study, performed by Rosa et al., showed that gene expression data obtained from lesional skin in the first 4 weeks of treatment with four specific drugs—Etanercept, Ustekinumab, Adalimumab, and Methotrexate—could predict clinical outcome by week 12. Through this, the clinical assessment gap could be reduced at least by two months [87]. One of the first GWAS studies evaluating treatment response variation (baseline and at weeks 4 and 12) in 65 psoriasis patients after anti-TNFα therapy found strong association of single nucleotide polymorphisms (SNPs) in JAG2 and ADRA2A, which were associated with treatment responses to anti-TNF-α agents [88].

A recent study by Tsoi et al. demonstrated that, by integrating transcriptomic data from RNA-seq and genomic data of in vitro cytokine responses to a cohort of etanercept treated psoriatic patients, effective drug response could be assessed [89]. Molecular profiles were assessed in more than 200 RNA-seq samples, which showed a correlation between drug response and molecular changes during the treatment. Additionally, the study also showed that non-lesional gene expression data was a better predictor of clinical response to etanercept compared to lesional psoriatic skin. Another study on the clinical response to treatment with etanercept showed that signals of response in patients could be detected at baseline (in lesional, non-lesional skin, and blood) even before commencing the therapy. Gene expression profiling, through RNA-seq and serum proteome analysis from blood, showed that there was a strong association between clinical response and TNF-regulated genes in skin and blood [90]. The use of a machine-learning algorithm to predict patients’ response to three of the psoriasis biologics was studied in a recent study [91]. In this large cohort study, involving 242 psoriasis patients, dermal patch biomarker patches were applied before and after 12 weeks of drug treatment with IL-23, IL-17, and TNF-α. By combining the dermal transcriptome with machine learning algorithm, it was found that the patient response rate was 64% for the entire cohort and 47.6–72.5% for different biologics. Only one patient did not respond to any of the treatment. Such studies effectively reduce the trial-and-error approach to the biologic treatment of psoriasis and emphasises the role pf precision medicine in the current management of psoriasis. Another similar study was performed using the blood samples of 266 moderate to severe psoriasis patients, at baseline and 4 weeks after treatment with Tofacitinib and Etanercept. Statistical and machine learning techniques were used to analyse this data to predict the treatment response after 12 weeks. Many relevant psoriasis markers including IL-17A and IL-17C were enriched in this blood transcriptome study. Such studies also highlight the importance of blood transcriptome profiles as another, less invasive platform to study drug response prediction, along with skin transcriptome [92]/\.

Thus, precision medicine, integrated with multi-omics techniques and analytics, could significantly improve the clinical response to the biological treatment of psoriasis in individual patients. This will allow physicians to identify patients who are likely to respond to a particular drug with higher efficacy and outcome thereby limiting unnecessary drug exposures.

## 6. Conclusions

In this review, we summarised the findings of several larger cohort transcriptional profiling studies and their importance in identifying the differentially expressed transcripts in psoriatic skin. Additionally, we highlighted the impact of considering the expression profile of alternatively spliced transcripts in psoriasis, rather than looking at the gene. Psoriasis is characterized by significant differences in the expression of RNA alternative isoforms, and gaining insight to these new isoforms is essential to design precise therapies for the treatment of psoriasis leading to a better treatment outcomes. One of the main limitations in the current treatment of psoriasis is the poor outcome of clinical response to the currently available drugs. Precision medicine using the vast amount of transcriptomic data is one method to reduce the assessment gap often seen during drug therapy. Such drug assessment approaches will help physicians to identify patients who will respond to a particular drug with maximum efficacy, as well as reduce the cost and time associated with the current management of psoriasis.

## Figures and Tables

**Figure 1 ijms-23-06130-f001:**
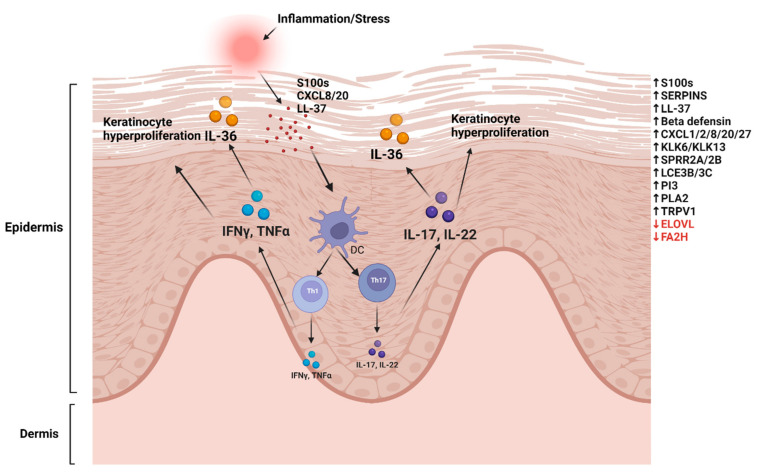
Molecular changes in psoriasis inflammation or cell stress induces the release of S100 proteins, AMPs, and cathelicidins. These activate dendritic cells (DCs) to induce T-cell proliferation to Th1 and Th17. IL-17/Il-22 induces epidermal (keratinocyte) proliferation, inflammatory cell infiltration, paraketosis, and acanthosis. IL17 and TNF-α upregulate IL-36 expression in inflamed skin. Gene marked in black represent upregulated genes, and red represents downregulated genes (as reviewed in the manuscript). The image was made by using Biorender (biorender.com).

**Table 1 ijms-23-06130-t001:** Large scale (>20 samples) psoriasis transcriptomic analyses.

Authors/Ref	Samples	Findings	Year
Oestreicher [15]	**24** psoriatic skin biopsies –lesional and uninvolved	159 DEGs were generated including S100A7, S100A12, elafin, KRT16, KRT17, MMP12, FARP5	2001
Nattkemper [34]	**25** patients with atopic dermatitis and 25 patients with psoriasis	18,000 DEGs common between itchy, lesional atopic, and psoriatic skin identified, outofwhich 2000 genes were unique to both AD and psoriasis including CCL-2/3/17/18, IL-6/8/17A/22/23A, S100A9/A15, TRPV1, PLA2	2007
Yao [47]	**26** paired nonlesional and lesional (all were plaque-type) skin biopsies from 26 psoriatic patients	Type 1 IFNS were significantly elevated in psoriatic lesions suggesting their active signaling in psoriasis	2008
Ainali [43]	**37** patients affected by chronic plaque psoriasis.	A comprehensive analysis of gene expression in paired lesional and non-lesional psoriatic tissue samples revealed different molecular subgroups associated with Wnt, Notch, TGF-beta, ErbB signaling pathways.	2012
Ewald [54]	Four microarray datasets including **54** LS and **43** NL samples	Differentially expressed in AD several genes involved in lipid metabolism including FA2H, critical in maintaining the permeability barrier of epidermis and ELOVL3, encoding a protein involved in the elongation of long chain fatty acids and essential in prevention of trans-epidermal water loss.	2015
Gudjonsson [41]	**58** psoriatic subjects	Uninvolved psoriatic skin exists in a a “prepsoriatic” gene expression signature and downregulation of PPARA, ESR2 and SREBBF1 suggesting decreased lipid biosynthesis and increased innate immunity in uninvolved psoriatic skin.	2009
Gudjonsson [37]	**58** psoriatic subjects	Identified over 600 new transcripts SERPINB4, PI3, DEFB4 and several S100 family members. Comparison of the psoriatic transcriptome to the transcriptomes of cytokine stimulated cultured keratinocytes revealed little overlap with the lesional psoriatic dysregulated transcriptome.	2010
Swindell [50]	**62** lesional skin samples obtained from patients with stable chronic plaque psoriasis.	Variability in cytokine signature was identified by whole genome transcriptional profiling	2012
Suez Farinas [46]	Skin biopsies from 85 paired lesional and non-lesional samples from a cohort of patients with moderate-to-severe psoriasis vulgaris who were not receiving active psoriasis therapy	Identified 2725 genes as being differentially expressed in psoriasis and link to functional pathways associated with metabolic diseases/diabetes and to cardiovascular risk pathways	2012
Li [31]	**92** psoriatic skin biopsies	RNA-seq analysis identified differentially expressed transcripts enriched for lymphoid and/or myeloid signature transcripts and genes induced by IL-17 in keratinocytes.	2014
Swindell [51]	**163** biopsies from psoriatic lesions	Identified 1233 psoriasis-increased DEGs attributing to keratinocyte activity, infiltration of lesions by T-cells, and macrophages (11%).	2013
Tian [53]	5 microarray data sets, including **193** LS and NL pairs	Several new genes were identified that are involved in cardiovascular development and lipid metabolism. highlighting the relationship between psoriasis and systemic manifestations such as the metabolic syndrome and cardiovascular disease	2012

## Data Availability

Not applicable.

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
