# Peer review of "Transcriptional Basis of Psoriasis from Large Scale Gene Expression Studies: The Importance of Moving towards a Precision Medicine Approach"

_ijms, 2022, doi:10.3390/ijms23116130_

Round 1
Reviewer 1 Report
Valuable and thoroughly designed review paper summarizing the results of several large scale psoriatic transcriptomic studies and their importance in indentyfing the differentialy expresses transcripts in psoriatic skin. I'd reccomend citing another paper when discussing the role of PCSK9 in psoriasis. Krahel et al. in Journal of Clinical Medicine published valuable research and pointed that PCSK9 might be a novel marker of psoriasis and its lipid disturbancies related but also evaluated its level with relation to systemic therapies (https://www.ncbi.nlm.nih.gov/pmc/articles/PMC7230388/).
Author Response
RESPONSE TO REVIEWER COMMENTS
We thank the referees for their constructive comments that have improved the quality of the revised manuscript. Our responses to specific issues raised by the reviewers (italicised) are detailed. All changes made to the manuscript are shown with track changes and additional references are highlighted in yellow
Reviewer 1:
Comments: Valuable and thoroughly designed review paper summarizing the results of several large scale psoriatic transcriptomic studies and their importance in identifying the differentially expressed transcripts in psoriatic skin. I'd reccomend citing another paper when discussing the role of PCSK9 in psoriasis. Krahel et al. in Journal of Clinical Medicine published valuable research and pointed that PCSK9 might be a novel marker of psoriasis and its lipid disturbancies related but also evaluated its level with relation to systemic therapies(https://www.ncbi.nlm.nih.gov/pmc/articles/PMC7230388/).
Thank you! We have now cited this paper in our review
Reviewer 2
Comments: This is a great review summarizing transcriptomics studies in the context of skin inflammatory disease- psoriasis. Authors have carefully cataloged various articles highlighting transcriptomics changes not only in psoriasis skin but also in blood from psoriasis patients. Furthermore, this article also discuss the importance of alternative splicing and mRNA isoforms in psoriasis. It is a well-conceived and nicely written review article. I have following few comments:
(i) What are authors view on precision medicine in psoriasis? Given the great success of biologics (anti-TNF, anti-IL-17 and anti-IL-23) in treating psoriasis, how the precision medicine approach would fair in individualizing psoriasis treatment. This should be highlighted, as the title of this review is also about precision medicine approach in psoriasis.
Thank you. We have now included a section “Precision medicine in Psoriasis management” in the manuscript.
(ii) Could author provide a graphical model of molecular changes in psoriasis based on the transcriptomics analysis reviewed in this manuscript? This would be greatly beneficial for the readers of this article.
Thank you. We have now included figure illustrating Molecular changes in psoriasis
(iii) There are some minor spelling mistakes, for example line 80, earlt would be early. Authors should scan the whole manuscript for such mistakes.
We have corrected this and scanned the entire manuscript for spelling mistakes
(iv) There are some punctuation mistakes, such as in line 175, there is period before and after the reference number 43. Authors should scan the whole manuscript for such errors.
We have corrected this. Thank you.
Reviewer 2 Report
This is a great review summarizing transcriptomics studies in the context of skin inflammatory disease- psoriasis. Authors have carefully cataloged various articles highlighting transcriptomics changes not only in psoriasis skin but also in blood from psoriasis patients. Furthermore, this article also discuss the importance of alternative splicing and mRNA isoforms in psoriasis. It is a well-conceived and nicely written review article. I have following few comments,
- What are authors view on precision medicine in psoriasis? Given the great success of biologics (anti-TNF, anti-IL-17 and anti-IL-23) in treating psoriasis, how the precision medicine approach would fair in individualizing psoriasis treatment. This should be highlighted, as the title of this review is also about precision medicine approach in psoriasis.
- Could author provide a graphical model of molecular changes in psoriasis based on the transcriptomics analysis reviewed in this manuscript? This would be greatly beneficial for the readers of this article.
- There are some minor spelling mistakes, for example line 80, earlt would be early. Authors should scan the whole manuscript for such mistakes.
- There are some punctuation mistakes, such as in line 175, there is period before and after the reference number 43. Authors should scan the whole manuscript for such errors.
Author Response

(The authors gave the same response as above.)
